# Diversity-Guided MLP Pruning for Efficient Large Vision Transformers

## Abstract

Transformer models achieve excellent scaling property, where the performance is improved with the increment of model capacity. However, large-scale model parameters lead to an unaffordable cost of computing and memory. We analyze popular transformer architectures and find that multilayer perceptron (MLP) modules take up the majority of model parameters.

To this end, we focus on the recoverability of the compressed models and propose a Diversity-Guided MLP Pruning (DGMP) method to significantly reduce the parameters of large vision transformers with only negligible performance degradation. Specifically, we conduct a Gram-Schmidt weight pruning strategy to eliminate redundant neurons of MLP hidden layer, while preserving weight diversity for better performance recover during distillation. Compared to the model trained from scratch, our pruned model only requires 0.06% data of LAION-2B (for the training of large vision transformers) without labels (ImageNet-1K) to recover the original performance. Experimental results on several state-of-the-art large vision transformers demonstrate that our method achieves a more than 57.0% parameter and FLOPs reduction in a near lossless manner. Notably, for EVA-CLIP-E (4.4B), our method accomplishes a 71.5% parameter and FLOPs reduction without performance degradation. The source code and trained weights will be publicly available.

## 1 Introduction

Transformer models have presented excellent scaling property on computer vision and nature language processing, where the performance of model can be consistently improved with the increment of model size. While remarkable performance is achieved by large transformer models, excessive computing and memory costs greatly challenge their economic deployment on widespread applications.

To this end, several model pruning (Li et al., 2017) and knowledge distillation (Hinton, 2015) methods are designed to compress cumbersome vision transformers (ViT) (Dosovitskiy, 2020). For pruning-based methods, the main idea focuses on the importance metrics of model weights (Yang et al., 2023; Zheng et al., 2022; Chavan et al., 2022) or attention heads (Shim et al., 2024; Wang et al., 2022; He & Zhou, 2024). The weights or attention heads with small importance scores are discarded to accelerate transformer models. However, these methods ignore the weight diversity of the pruned model, which may miss vital information to recover the performance of the original model. Moreover, gradient-based pruning methods (Yang et al., 2023; Zheng et al., 2022; Wang et al., 2022) require massive gradient computations and inefficient iterative pruning-finetuning round, which significantly increase the cost of model compression, particularly for large vision transformers. Distillation methods (Hinton, 2015; Oquab et al., 2023) follow a teacher-student paradigm, where the pretrained large model is regarded as teacher to guide the training of a lightweight student model. Due to inconsistent model structure between the student and the teacher, the student model is generally trained from scratch, thus requiring long training schedule and large computing cost on large dataset to recover the performance, especially for large vision transformers.

Due to large expansion ratio of hidden layer, the parameters of MLP modules in Transformer architecture are dominant part across the whole model. For example, MLP modules of EVA-CLIP-E (Sun et al., 2023) have an 81.1% parameter share of the whole model. As shown in Figure 1(a), the input

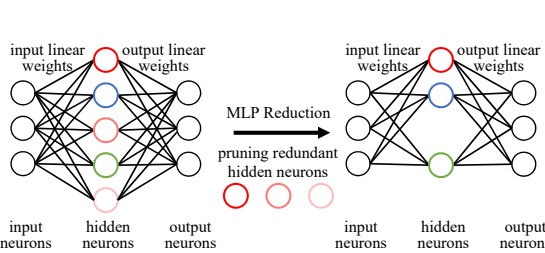

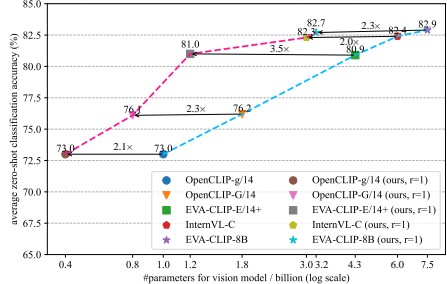

(a) MLP pruning for efficient large vision transformers.

(b) Average zero-shot accuracy on ImageNet variants and ObjectNet.

Figure 1: (a) Due to large MLP expansion ratio of vision transformer models, there exist large amounts of redundant parameters for model compression. (b) Our method achieves encouraging performance on model compression of the existing state-of-the-art large transformer models in a near lossless manner.

neurons are combined to form the hidden neurons with a significant expansion, whose value varies from 2.67 (DINOv2-g) to 8.57 (EVA-CLIP-E) in practice. During the training of transformer, large expansion ratio can well benefit the convergence of the trained model (Frankle & Carbin, 2019), but introduces massive redundant parameters in MLP modules. Therefore, the reduction of MLP's hidden layer can effectively compress large vision transformer model.

In this paper, we focus on the elimination of redundant neurons in MLP modules of large vision transformers. Our core insight is that the redundant neurons can be well replaced by the combination of several principal neurons after proper parameter finetuning. The key issues are that how to determine the principal hidden neurons of MLP module and how to recover the original performance.

To this end, we propose a novel Diversity-Guided MLP Pruning (DGMP) method, which aims to filter important neurons from MLP module without loss of neuron diversity. First, we evaluate the importance of neurons by the strength of connection (e.g. weight magnitude) between input neurons and hidden neurons, and then choose the hidden neuron (or weight) with the largest importance score. Afterward, we eliminate the weights of previous selected neuron components from the weights of the rest neurons and obtain updated weights, thus avoiding overlapping with previous selected neurons. Then, we repeat the first step to obtain the second most important neuron and add it into the selected neuron set. In this manner, the follow-up selected neurons embrace the most information, which isn't covered by previous selected neurons, thereby maximizing the diversity of the selected neurons. As the above procedure, the neurons are selected in the order of importance score, until the target size of pruned model is reached. These selected neurons are regarded as the principal ones, whose combinations can be used to replace the redundant neurons. Moreover, our pruning method doesn't require additional gradient computations or iterative pruning-finetuning round.

To recover the performance of the above pruned model, we further conduct knowledge distillation, where the original model serves as the teacher of the pruned model. Owing to the affinity between the original model and the pruned one, the performance of the pruned model can be effectively recovered.

To validate our proposed method, we evaluate it on various state-of-the-art large vision transformer models, including EVA-CLIP-E (Sun et al., 2023), EVA-CLIP-8B (Sun et al., 2024) and DINOv2-g (Oquab et al., 2023). With only distillation on ImageNet-1K, our method achieves a 71.5% parameter (and FLOPs) reduction on EVA-CLIP-E, a 57.0% parameter (and FLOPs) reduction on EVA-CLIP-8B and a 59.3% parameter (and FLOPs) reduction on DINOv2-g in a near lossless manner as Figure 1(b).

The main contributions of our method can be summarized as follows.

- We explore a lossless model compression technique for large vision transformer models without computing-intensive iterative pruning-finetuning procedure.

- We propose a diversity-guided MLP pruning method, which effectively preserves the diversity of pruned model, thus improving the recoverability of the pruned model.

- With only distillation on ImageNet-1K, our method achieves a more than 57.0% parameter and FLOPs reduction on state-of-the-art large vision transformers with only negligible performance loss, even outperforming the original models in some cases.

## 2 RELATED WORK

### 2.1 MODEL PRUNING FOR VISION TRANSFORMERS

To accelerate the inference and reduce the memory consumption for ViT, several works (Chen et al., 2021; Yu et al., 2022) are developed to compress the size of multi-head self-attention modules or multilayer perceptron modules. A general idea of these methods is to measure the importance scores of model weights, and then prune the least important weights for model compression.

Magnitude-based pruning methods (Han et al., 2015; Li et al., 2017; Liu et al., 2017) evaluate the importance score of weights by the magnitude of the corresponding weights, where the weight with larger magnitude are regarded as the more important one in the model. ViT-Slim (Chavan et al., 2022) applies a learnable $\ell_1$ sparsity constraint as the global importance score to guide the ViT sub-structure search for efficient architecture. DIMAP (He & Zhou, 2024) analyzes the contribution of local weights by their information distortion to reduce the case, where some important locally important weights are pruned by mistake.

Attention-based pruning methods (Guo et al., 2024) determine informative weights according to attention scores from multi-head self-attention modules. SNP (Shim et al., 2024) prunes graphically connected query and key layers with the least informative attention scores, while keeping the overall attention scores. Hence, the pruned query and key layers have a smaller impact on the final predictions of the model.

Taylor-based pruning methods (Molchanov et al., 2017; 2019) introduce a novel importance criterion based on Taylor expansion to approximate the change of loss function caused by the pruned parameters, thus minimizing the effect of model pruning. SAViT (Zheng et al., 2022) adopts a Taylor-based approximation to evaluate the joint importance across all components to perform collaborative pruning for a more balanced parameter reduction. VTC-LFC (Wang et al., 2022) introduces low-frequency sensitivity metric to estimate importance score of model weights and approximates this metric with first-order Taylor expansion to conduct model pruning. NViT (Yang et al., 2023) introduces Hessian-based pruning criterion across transformer blocks to determine the importance of the group of parameters for a global model pruning. The Hessian-based criterion proposed by NViT shares a similar formula as Taylor-based pruning criterion.

The above methods mainly aim to minimize the effect on the predictions of the pruned model. In contrast, our method preserves weight diversity of the pruned model, thus maximizing the performance recover during distillation. Moreover, different from the above pruning methods, our method doesn't require computing-intensive iterative pruning or additional gradient computation, thereby more friendly to the model compression of large vision transformers.

### 2.2 TOKEN REDUCTION FOR VISION TRANSFORMERS

Besides pruning model parameters, another idea to accelerate model inference is reducing the number of tokens fed into ViT. The flexible solutions for token reduction include token pruning and token merging.

Token pruning methods aim to prune unimportant tokens of sequence for efficient inference. A-ViT (Yin et al., 2022) dynamically prunes tokens and preserve discriminative tokens at different depths. AdaViT (Meng et al., 2022) adaptively applies patches, heads and layers of ViT conditioned on input images for efficient inference. DynamicViT (Rao et al., 2021) adopts an attention masking strategy to differentiably prune a token by blocking its interactions with other tokens. LRP (Luo et al., 2024) computes semantic density score of each patch by quantifying variation between reconstructions with and without this patch to rank the patches for token pruning. Zero-TPrune (Wang et al., 2024) leverages attention graph of pretrained Transformers to produce an importance distribution for efficient similarity-based token pruning.

Token merging methods fuse several tokens into one for token reduction. ToMe (Bolya et al., 2023) adopts a bipartite soft matching algorithm to fast merge the most similar tokens, thus reducing the number of tokens for inference. BAT (Long et al., 2023) distinguishes attentive and inattentive tokens according to class token, then merge similar inattentive tokens and match attentive token to maximize the token diversity. TPS (Wei et al., 2023) applies unidirectional nearest-neighbor matching and similarity-based fusing steps to squeeze the information of pruned tokens into partial reserved tokens for lossless model compression. STViT (Chang et al., 2023) introduces semantic tokens to represent cluster centers of the whole tokens for global or local semantic information. TokenLearner (Ryoo et al., 2021) learns to compute important regions of image/video as spatial attention for the tokenization. Vid-TLDR (Choi et al., 2024) captures the salient regions in videos with attention map and then conduct saliency-aware token merging by dropping the background tokens and sharpening the object scores.

Our method focuses on the parameter reduction of large vision transformer model, while compatible with the existing token reduction methods. The combination of our parameter reduction method and the above token reduction methods can further accelerate the inference of transformer models.

# 3 METHOD

## 3.1 OVERVIEW

The goal of model compression is to maximize the reduction of model size, while minimizing the performance drop of the compressed model. To accomplish this goal on large vision transformer models, we pay attention to the parameter reduction of MLP modules, which contain most of the parameters of transformer architecture. Meanwhile, we consider the diversity of weight after model pruning to reduce the information loss relative to the original model.

As depicted in Figure 2, we adopt a dual stage model compression strategy to reduce the model size in a lossless manner. In the first stage, we prune the hidden layer of parameter-intensive MLP modules, while preserving the diversity of weighted connections. After pruning, the feature dimension of token keeps consistent with the original one, thus minimizing the effect to other modules. In the second stage, we conduct knowledge distillation scheme to guide the pruned model to recover the performance of the original model, where the original model is regarded as the teacher. Thanks to the identical output dimensions after pruning, the alignment between the teacher and the student can be seamlessly implemented without additional modules.

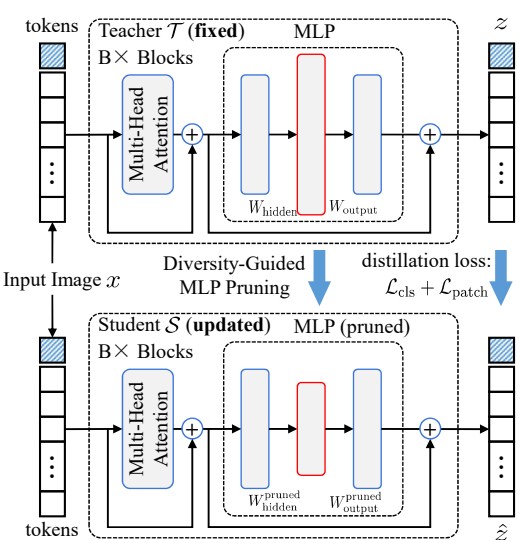

Figure 2: The overview of the proposed method. In the first stage, we prune the hidden neurons of MLP modules in a diversity preserving manner, where the MLP modules contain most of the parameters of transformer model. In the second stage, the original transformer model works as the teacher to guide the training of the above pruned model for performance recover.

## 3.2 DIVERSITY-GUIDED MLP PRUNING

In this section, we focus on the reduction of parameter-intensive MLP modules, to compress large vision transformer models. Our main target is to reduce redundant information but preserve weight or neuron diversity after pruning, thus improving the recoverability of the pruned model.

As depicted in Figure 2, we present weights of MLP hidden layer as $W_{\text{hidden}} = [w_1, w_2, \cdots, w_M]^\top \in \mathbb{R}^{M \times N}$, where $M$ and $N$ denote the number of hidden neurons and the one of input neurons, respectively; $w_i \in \mathbb{R}^N$ denotes the weight of $i$-th neuron. The bias of hidden

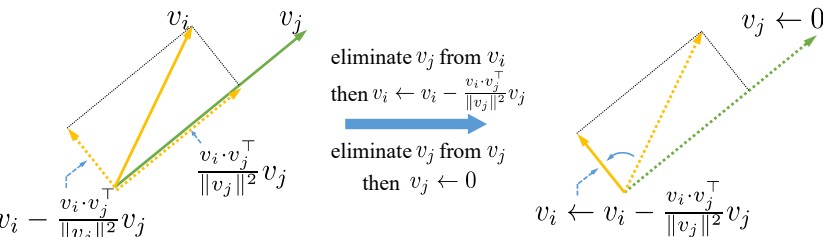

Figure 3: We eliminate $v_j$ of neuron $j$ from $\{v_i\}_i$. The neuron with the largest $\ell_2$ norm of $\{v_i\}_i$ obtained by Gram-Schmid algorithm is selected in the next round. Hence, the next selected neuron contains the most information missed by the previous ones.

layer can be denoted as $b_{\text{hidden}} \in \mathbb{R}^M$. $W_{\text{output}} \in \mathbb{R}^{N \times M}$ indicates the weights of MLP output layer.

To evaluate the importance of hidden neurons, we adopt the magnitude of weight connection between hidden neuron and input neurons as neuron importance criterion. First, we set $V = [v_1, v_2, \cdots, v_M]$, where $v_i$ is initialized with $w_i$. Then, we select the neuron with the largest $\ell_2$ norm according to

$$j = \arg\max i \{\|v_i\|\}_{i=1}^M, \tag{1}$$

and add the selected neuron $j$ into set $\mathcal{J}$.

To maximize the diversity of selected neurons in set $\mathcal{J}$, we apply Gram-Schmid algorithm on the rest neuron weights $\{v_i\}_{i=1}^M$ as Figure 3 by

$$v_i \leftarrow v_i - \frac{v_i \cdot v_j^\top}{\|v_j\|^2} v_j, \tag{2}$$

where the weight component of the selected neuron $v_j$, namely $\frac{v_i \cdot v_j^\top}{\|v_j\|^2} v_j$, is eliminated from the weights of the rest neurons $\{v_i\}_{i=1}^M$. Note that the weight of the selected neuron $v_j$ equals to zero after elimination and will not be selected by the metric of the largest $\ell_2$ norm in the next round, thereby not affecting the selection of later neurons.

Then, we select the next neuron by Table 1 and add it into set $\mathcal{J}$. We repeat the above steps until the target size of hidden layer is reached. We set the target expansion ratio of pruned MLP as $r = \frac{M_{\text{pruned}}}{N}$, where $M_{\text{pruned}}$ denotes the hidden size of MLP after pruning. In this manner, the neurons with the most information, which the previous selected neurons don't contain, are preserved, thus improving the weight diversity of selected neurons.

For hidden layer of MLP, we follow a structural pruning strategy and the final pruned weight and bias can be obtained by

$$W_{\text{hidden}}^{\text{pruned}} = W_{\text{hidden}}[\mathcal{J}, :], \tag{3}$$
$$b_{\text{hidden}}^{\text{pruned}} = b_{\text{hidden}}[\mathcal{J}].$$

Based on the selected neurons after structural pruning, the pruned weight of output layer can be obtained by

$$W_{\text{output}}^{\text{pruned}} = W_{\text{output}}[:, \mathcal{J}]. \tag{4}$$

Note that $b_{\text{output}}$ isn't required to be pruned, since the dimension of output remains unchanged. The overall algorithm can be found in the appendix.

### 3.3 KNOWLEDGE DISTILLATION FOR RECOVER

In Sec. 3.2, diversity-guided MLP pruning aims to preserve the knowledge of the original model without an obvious change of model structure when pruning the model. Due to weight and structure affinity between the original transformer model and the pruned model, we take the original model as the teacher to guide the performance recover of the pruned model.

As Figure 2, the image $x$ is fed into both teacher $\mathcal{T}$ and student $\mathcal{S}$ to obtain the output of the last transformer block as $[z_{\text{cls}}, z_{\text{patch}}] = \mathcal{T}(x)$ and $[\hat{z}_{\text{cls}}, \hat{z}_{\text{patch}}] = \mathcal{S}(x)$, where $z_{\text{cls}} \in \mathbb{R}^C$ and

$z_{\text{patch}} \in \mathbb{R}^{L \times C}$ respectively denote the representations of class token and patch tokens; $L$ and $C$ stand for the length of image patch sequence and the dimension size of single image token, respectively. The dimension sizes of $\hat{z}_{\text{cls}}$ and $\hat{z}_{\text{patch}}$ are respectively identical to the ones of $z_{\text{cls}}$ and $z_{\text{patch}}$.

The representation of class token $z_{\text{cls}}$ is generally used for prediction, while the ones of patch tokens $z_{\text{patch}}$ are derived from input image data. To balance the capability of prediction and data fitting, we conduct knowledge distillation using mean squared error loss for both class token and patch tokens as

$$\mathcal{L} = \mathcal{L}_{\text{cls}} + \mathcal{L}_{\text{patch}} = \frac{1}{C} \|z_{\text{cls}} - \hat{z}_{\text{cls}}\|^2 + \frac{1}{L \cdot C} \|z_{\text{patch}} - \hat{z}_{\text{patch}}\|^2. \tag{5}$$

## 4 EXPERIMENTS

### 4.1 EXPERIMENTAL SETTINGS

To recover the performance of the pruned models, we implement knowledge distillation on the pruned models using ImageNet-1K (Russakovsky et al., 2015) without labels, which contains only 0.06% data of LAION-2B (Schuhmann et al., 2022) used to train the original models. All images used for distillation and evaluation are resized into $224 \times 224$. To validate the effectiveness of our proposed method, we evaluate our method on several popular benchmarks. For CLIP-style models, we evaluate the distilled vision models on zero-shot image classification of various ImageNet variants (including the validation set of ImageNet-1K, ImageNet-V2 (Recht et al., 2019), ImageNet-Adv (Hendrycks et al., 2021b), ImageNet-R (Hendrycks et al., 2021a) and ImageNet-Sketch (Wang et al., 2019)) and ObjectNet (Barbu et al., 2019) using CLIP benchmark (LAION-AI, 2023). To further substantiate the superiority of our method, we also estimate the distilled model on zero-shot image and text retrieval tasks of Flickr30K (Young et al., 2014) and COCO (Lin et al., 2014) using CLIP benchmark.

For pure vision model, such as DINOv2-g, we evaluate the performance of the distilled model on ImageNet-1K using kNN evaluation protocol. For more comprehensive comparison, we also evaluate CLIP-style models using kNN evaluation protocol. To evaluate the generality of our method, we further assert our method on another Transformer variant, Swin Transformer architecture (Liu et al., 2021) in the settings of supervised image classification task (see the appendix).

For model pruning, we prune MLP modules of all vision transformer models as expansion ratio $r = 1$ and $r = 2$. In other words, the hidden dimension size of MLP is pruned as identical size as token dimension size when $r = 1$, and twice of token dimension size when $r = 2$. For CLIP-style models, only vision transformer model is compressed and the corresponding text encoder remains unchanged.

For knowledge distillation, all models are trained on servers with $8\times$ A6000 GPUs for 10 epochs, where the first 1 epoch is used for learning rate warming-up. We distill all models using AdamW (Loshchilov, 2017) optimizer with bfloat16 precision. The learning rates for all models follow a cosine schedule from lr to zero, where lr = base_lr $\times$ batch_size / 256. The base learning rates and batch sizes for different vision transformer models are listed in the supplementary material.

The models, including OpenCLIP-g (Cherti et al., 2023), OpenCLIP-G (Wortsman, 2023) and EVA-CLIP-E (Sun et al., 2023), DINOv2-g (Oquab et al., 2023), are trained by distributed data parallel (DDP) strategy. The models with more than 6 billion parameters, including InternVL-C (Chen et al., 2024) and EVA-CLIP-8B (Sun et al., 2024), are trained by fully sharded data parallel (FSDP) strategy. For all models, the patch size for embedding is set to 14. More implementation details can be found in the appendix.

### 4.2 ZERO-SHOT IMAGE CLASSIFICATION

To validate the effectiveness of our method, we compress state-of-art CLIP-style models as MLP expansion ratio $r = 1$ and $r = 2$, then evaluate their performance on zero-shot image classification task of various ImageNet variants and ObjectNet.

Table 1: The performance comparison on zero-shot image classification tasks of various ImageNet variants and ObjectNet. Note that InternVL-C adopts an obviously larger text encoder (8B parameters), QLLaMA, than other CLIP-style models; $r$ denotes MLP expansion ratio after pruning. "#Params" denotes the number of vision encoder's parameters, excluding the parameters of text encoder. Throughout is averaged over 8 runs on single A6000 GPU with batch size of 256.

| Method | #Params | FLOPs | Throughout | IN-1K | IN-Adv | IN-R | IN-V2 | IN-Ske | ObjectNet | Avg. Acc. |
|---|---|---|---|---|---|---|---|---|---|---|
| OpenCLIP-g (Cherti et al., 2023) | 1.01B | 0.52T | 161.6 imgs/s | 78.5% | **60.8%** | 90.2% | 71.7% | 67.5% | 69.2% | 73.0% |
| OpenCLIP-g (ours, $r=1$) | **0.48B** | **0.25T** | **272.1 imgs/s** | 78.5% | 60.4% | **90.3%** | 71.7% | **67.9%** | 69.2% | 73.0% |
| OpenCLIP-g (ours, $r=2$) | 0.64B | 0.33T | 225.7 imgs/s | **78.6%** | 60.6% | **90.3%** | **71.9%** | **67.9%** | **69.6%** | **73.2%** |
| OpenCLIP-G (Wortsman, 2023) | 1.84B | 0.95T | 95.2 imgs/s | **80.1%** | **69.3%** | 92.1% | **73.6%** | 68.9% | 73.0% | **76.2%** |
| OpenCLIP-G (ours, $r=1$) | **0.80B** | **0.41T** | **177.0 imgs/s** | **80.1%** | 69.1% | 92.2% | 73.5% | 68.9% | 73.0% | 76.1% |
| OpenCLIP-G (ours, $r=2$) | 1.07B | 0.55T | 144.9 imgs/s | 80.0% | 68.5% | **92.2%** | **73.6%** | **69.2%** | 73.0% | 76.1% |
| EVA-CLIP-E (Sun et al., 2023) | 4.35B | 2.23T | 46.0 imgs/s | **82.0%** | 82.1% | 94.5% | 75.7% | 71.6% | 79.6% | 80.9% |
| EVA-CLIP-E (ours, $r=1$) | **1.24B** | **0.63T** | **139.9 imgs/s** | 81.9% | **82.3%** | 94.6% | 75.8% | 71.8% | 79.5% | 81.0% |
| EVA-CLIP-E (ours, $r=2$) | 1.65B | 0.85T | 109.9 imgs/s | 81.9% | 82.2% | **94.6%** | **75.8%** | **72.0%** | **79.7%** | **81.1%** |
| InternVL-C (Chen et al., 2024) | 5.90B | 3.03T | 27.2 imgs/s | **83.2%** | 83.8% | 95.5% | 77.3% | 73.8% | 80.6% | **82.4%** |
| InternVL-C (ours, $r=1$) | **2.95B** | **1.52T** | **42.9 imgs/s** | 83.1% | 83.8% | 95.5% | 77.2% | 73.8% | 80.6% | 82.3% |
| InternVL-C (ours, $r=2$) | 3.93B | 2.02T | 36.1 imgs/s | **83.2%** | 83.8% | **95.6%** | **77.4%** | **73.9%** | 80.6% | **82.4%** |
| EVA-CLIP-8B (Sun et al., 2024) | 7.53B | 3.86T | 26.6 imgs/s | **83.5%** | 85.2% | **95.3%** | 77.7% | **74.3%** | 81.2% | **82.9%** |
| EVA-CLIP-8B (ours, $r=1$) | **3.23B** | **1.66T** | **54.2 imgs/s** | 83.1% | 85.1% | 95.1% | **77.8%** | 73.9% | 81.3% | 82.7% |
| EVA-CLIP-8B (ours, $r=2$) | 4.30B | 2.21T | 43.2 imgs/s | 83.4% | **85.8%** | **95.3%** | 77.7% | 74.1% | **81.4%** | **82.9%** |

Table 2: The performance comparison on zero-shot retrieval task of Flickr30K and COCO datasets. "R@K" is short for Recall@K. "MR" is short for mean recall, which is average value of all recall metrics.

| Method | #Params | Zero-Shot Text Retrieval (text → image) | | | | | | Zero-Shot Image Retrieval (image → text) | | | | | | MR |
|---|---|---|---|---|---|---|---|---|---|---|---|---|---|---|
| | | Flickr30K | | | COCO | | | Flickr30K | | | COCO | | | |
| | | R@1 | R@5 | R@10 | R@1 | R@5 | R@10 | R@1 | R@5 | R@10 | R@1 | R@5 | R@10 | |
| OpenCLIP-g (Cherti et al., 2023) | 1.01B | 91.5 | 98.9 | 99.5 | **66.4** | 86.0 | 91.8 | 77.5 | **94.1** | 96.7 | 48.7 | 73.2 | 81.4 | 83.8 |
| OpenCLIP-g (ours, $r=1$) | **0.48B** | 91.2 | 98.9 | 99.5 | 65.1 | 86.0 | 91.7 | 77.8 | 93.6 | 96.7 | **49.1** | **73.9** | 81.9 | 83.8 |
| OpenCLIP-g (ours, $r=2$) | 0.64B | **92.0** | 98.9 | **99.7** | 66.1 | **86.6** | **91.9** | **77.9** | **94.1** | 96.7 | **49.1** | 73.8 | **82.1** | **84.1** |
| OpenCLIP-G (Wortsman, 2023) | 1.84B | **92.6** | **99.4** | 99.8 | 66.9 | 87.2 | 92.8 | 79.8 | **95.1** | 97.0 | **51.3** | 74.8 | 82.9 | **85.0** |
| OpenCLIP-G (ours, $r=1$) | **0.80B** | 91.4 | 99.2 | **99.9** | 66.2 | 87.0 | 92.5 | 79.3 | 94.8 | 97.0 | 50.7 | 75.1 | 83.0 | 84.7 |
| OpenCLIP-G (ours, $r=2$) | 1.07B | 91.6 | 99.3 | **99.9** | 66.8 | **87.4** | **92.9** | 79.8 | **95.1** | 97.0 | 51.2 | **75.3** | **83.2** | **85.0** |
| EVA-CLIP-E (Sun et al., 2023) | 4.35B | **94.9** | 99.3 | **99.7** | **68.8** | 87.8 | 93.0 | 78.9 | 94.4 | **97.1** | 51.0 | 74.8 | 82.7 | 85.2 |
| EVA-CLIP-E (ours, $r=1$) | **1.24B** | 93.2 | 99.3 | **99.7** | 68.1 | 87.8 | **93.1** | 79.5 | 94.8 | 96.9 | **51.6** | 75.2 | 82.9 | 85.2 |
| EVA-CLIP-E (ours, $r=2$) | 1.65B | 93.4 | **99.4** | 99.6 | 68.5 | **87.9** | **93.1** | **79.8** | **94.9** | 97.0 | **51.6** | **75.3** | **83.1** | **85.3** |
| InternVL-C (Chen et al., 2024) | 5.90B | **93.8** | 99.7 | 100.0 | **70.3** | 89.2 | 93.8 | 82.1 | 96.0 | 98.1 | 54.1 | 77.1 | 84.8 | 86.6 |
| InternVL-C (ours, $r=1$) | **2.95B** | 93.5 | 99.7 | 100.0 | **70.3** | 89.2 | 93.8 | 82.1 | 96.0 | 98.1 | 54.1 | 77.1 | 84.8 | 86.6 |
| InternVL-C (ours, $r=2$) | 3.93B | 93.6 | **99.8** | 100.0 | 70.2 | 89.2 | **93.9** | **82.3** | **96.3** | **98.2** | **54.4** | **77.3** | **84.9** | **86.7** |
| EVA-CLIP-8B (Sun et al., 2024) | 7.53B | **94.4** | 99.4 | **99.7** | 69.6 | **88.6** | **93.2** | 80.9 | 95.3 | 97.4 | 51.7 | 75.0 | 82.7 | 85.7 |
| EVA-CLIP-8B (ours, $r=1$) | **3.23B** | 93.5 | 99.4 | **99.7** | 69.7 | **88.6** | 93.1 | 81.2 | 95.5 | 97.6 | 51.9 | 75.3 | 83.2 | 85.7 |
| EVA-CLIP-8B (ours, $r=2$) | 4.30B | 93.8 | **99.6** | **99.7** | 69.9 | 88.1 | 93.1 | **81.7** | **95.6** | **97.5** | **52.3** | **75.8** | **83.5** | **85.9** |

As Table 1, our compressed models consistently achieve comparable average zero-shot classification accuracy as the corresponding original models, while both of parameters and FLOPs of pruned models are reduced below 50% of the original ones. Moreover, the image throughout of our pruned model is also significantly improved. Especially for EVA-CLIP-E, the number of parameters is reduced from 4.35 billion to 1.24 billion, a 71.5% parameter reduction. Its FLOPs is also reduced by 71.5% over the original one. The image throughout of EVA-CLIP-E ($r=1$) accomplishes $3.0\times$ acceleration relative to the original EVA-CLIP-E model. When the expansion ratio $r=2$, the pruned EVA-CLIP-E even outperforms the original model by 0.2% average zero-shot classification accuracy. The above results demonstrate that our method can significantly compress the vision transformer models in a near lossless manner on zero-shot classification task.

Additionally, our pruned OpenCLIP-G ($r=1$) with 0.80 billion parameters is significantly superior to OpenCLIP-g with 1.01 billion parameters by 3.1% average zero-shot classification accuracy. The pruned EVA-CLIP-E ($r=1$) with 1.24 billion parameters outperforms OpenCLIP-G with 1.84 billion parameters by a large margin, 4.9% average zero-shot classification accuracy. Overall, our pruned models achieve better performance than the counterparts with comparable, even more parameters, which supports the effectiveness of our method.

## 4.3 ZERO-SHOT RETRIEVAL

We report the results on zero-shot text and image retrieval task of Flickr30K and COCO in Figure 2, where zero-shot text retrieval takes the given text as the query to search the corresponding image and zero-shot image retrieval takes the given image as the query to search the matching text description.

The experimental results show that our pruned model achieves comparable retrieval performance on mean recall metric as the original models for all vision transformer models. We find that the pruned models incline to obtain better performance on zero-shot image retrieval task, while the results on zero-shot text retrieval are slightly lower than the original ones. We analyze the potential reasons as follows. In our method, the vision encoder is finetuned after pruning, but the corresponding text

encoder is fixed and not adapted to the new compressed vision encoder. Therefore, the outputs of text encoder are not optimal for zero-shot text retrieval task with new compressed vision encoder.

To this end, we finetune the text encoder to adapt the new pruned vision encoder on the training set of Flickr30K. The results are reported in the appendix and demonstrate that our pruned vision transformers with the corresponding adopted text encoder can achieve comparable performance as the original ones, even outperforming the original ones in some cases. In summary, the model compressed by our method achieves comparable retrieval performance as the original model using significantly fewer parameters and FLOPs. It indeed supports the effectiveness of our method on the model compression of large vision transformers.

### 4.4 COMPARISON TO OTHER PRUNING METHODS

To substantiate the effectiveness of our pruning strategy, we compare our method with several popular pruning strategies, including random pruning, $\ell_2$ norm pruning and Taylor pruning based methods. For fair comparison, all methods only prune MLP modules of the given models, whose target expansion ratio $r = 1$. For random pruning, we randomly preserve the weights of MLP module and prune the given model into the target model size.

As the results in Table 3, we find that even random pruning strategy can recover partial performance of the original model. This result indeed reveals that there exists large amounts of redundant parameters in MLP modules of large vision transformers. Furthermore,

Table 3: The effect of pruning strategy on average zero-shot classification accuracy of ImageNet variants and ObjectNet. "original" denotes the original model without compression. The expansion ratios of all models after pruning are set to $r = 1$. All pruned models are trained using knowledge distillation.

| Method | OpenCLIP-g | EVA-CLIP-E |
|---|---|---|
| original | 73.0% | 81.0% |
| random pruning | 70.8% | 78.2% |
| $\ell_2$ norm pruning | 71.4% | 79.3% |
| Taylor pruning (Molchanov et al., 2017) | 71.3% | 79.1% |
| SAViT (Zheng et al., 2022) | 71.4% | 79.2% |
| NViT (Yang et al., 2023) | 71.5% | 79.3% |
| DGMP (ours, $r = 1$) | **73.0%** | **81.0%** |

our method is obviously superior to other pruning methods. For EVA-CLIP-E, our method outperforms the second best $\ell_2$ norm strategy by 1.7% average zero-shot accuracy. It supports that our pruning strategy can well preserve the weight diversity after pruning and efficiently improve the recoverability of the pruned models. Moreover, compared to Taylor pruning, our method doesn't require time-consuming iterative pruning, training data or gradient computation, thus more applicable to the compression of large vision transformer models.

### 4.5 kNN EVALUATION

For more comprehensive comparison, we further evaluate the pruned vision transformers on pure vision task of ImageNet-1K without the involvement of text encoder using kNN evaluation protocol.

As depicted in Table 4, the models pruned by our method with $r = 1$ achieve similar kNN accuracy as the corresponding models. For example, kNN accuracy of the pruned OpenCLIP-g ($r = 1$) is 81.9%, slightly surpasses the one of the original OpenCLIP-g model by 0.2%. In spite of obviously fewer parameters and FLOPs, the pruned models with $r = 2$ consistently outperform the original models. Especially for OpenCLIP-g ($r = 1$), its performance gain over the original OpenCLIP-g model reaches 0.4% kNN accuracy. Moreover,

Table 4: The performance comparison on ImageNet-1K using kNN evaluation protocol. The outputs of the last transformer block are used to evaluate the kNN performance.

| Method | #Params | FLOPs | kNN (%) |
|---|---|---|---|
| OpenCLIP-g (Cherti et al., 2023) | 1.01B | 0.52T | 81.7 |
| OpenCLIP-g (ours, $r = 1$) | **0.48B** | **0.25T** | 81.9 |
| OpenCLIP-g (ours, $r = 2$) | 0.64B | 0.25T | **82.1** |
| OpenCLIP-G (Wortsman, 2023) | 1.84B | 0.95T | 82.9 |
| OpenCLIP-G (ours, $r = 1$) | **0.80B** | **0.41T** | 82.8 |
| OpenCLIP-G (ours, $r = 2$) | 1.07B | 0.55T | **83.1** |
| EVA-CLIP-E (Sun et al., 2023) | 4.35B | 2.23T | **85.8** |
| EVA-CLIP-E (ours, $r = 1$) | **1.24B** | **0.63T** | 85.7 |
| EVA-CLIP-E (ours, $r = 2$) | 1.65B | 0.85T | **85.8** |
| InternVL-C (Chen et al., 2024) | 5.90B | 3.03T | 85.2 |
| InternVL-C (ours, $r = 1$) | **2.95B** | **1.52T** | 85.2 |
| InternVL-C (ours, $r = 2$) | 3.93B | 2.02T | **85.3** |
| EVA-CLIP-8B (Sun et al., 2024) | 7.53B | 3.86T | **86.0** |
| EVA-CLIP-8B (ours, $r = 1$) | **3.23B** | **0.83T** | 85.9 |
| EVA-CLIP-8B (ours, $r = 2$) | 4.30B | 2.21T | **86.0** |
| DINOv2-g (Oquab et al., 2023) | 1.14B | 0.30T | 83.5 |
| DINOv2-g (ours, $r = 1$) | **0.66B** | **0.18T** | 83.5 |

our pruned models also achieve significantly superior performance against the models with comparable parameters. Despite fewer parameters of the pruned model, OpenCLIP-G ($r = 1$) with 0.80 billion parameters significantly outperforms OpenCLIP-g with 1.01 billion parameters by 1.1% kNN accuracy. Similarly, EVA-CLIP-E ($r = 1$) with 1.24 billion parameters outperforms OpenCLIP-G with 1.84 billion parameters by 2.8% kNN accuracy. To validate the generalization of our method,

we also evaluate our method on pure vision transformer model, DINOv2-g. The results also support that our method can effectively reduce the model size in a near lossless manner.

## 4.6 ABLATION STUDY

### 4.6.1 THE EFFECT OF MLP EXPANSION RATIO

In this section, we explore the effectiveness of different MLP expansion ratios for the compression of large vision transformers. The results are depicted in Table 5. The OpenCLIP-g model pruned by our DGMP with $r = 1$ achieves the similar average zero-shot classification accuracy on five ImageNet variants and ObjectNet as the original

Table 5: The effect of MLP pruning ratios on average zero-shot classification accuracy. "original" denotes the original model without compression.

| Method | OpenCLIP-g | | EVA-CLIP-E | |
|---|---|---|---|---|
| | #Params | Acc (%) | #Params | Acc (%) |
| original | 1.01B | 73.0 | 4.35B | 80.9 |
| DGMP ($r = 0.5$) | **0.40B** | 72.8 | **1.04B** | 80.5 |
| DGMP ($r = 1$) | 0.48B | 73.0 | 1.24B | 81.0 |
| DGMP ($r = 2$) | 0.64B | **73.2** | 1.65B | **81.1** |

OpenCLIP-g, while only leveraging 48.0% parameters of the original model. When the MLP expansion ratio $r$ reaches 2, our pruned model even outperforms the original OpenCLIP-g by 0.2% average zero-shot accuracy with only 64.0% parameters of the uncompressed version. We further study smaller expansion ratio as $r = 0.5$ for higher compression ratio. As Table 5, our pruned model with $r = 0.5$ achieves a 60.0% parameter reduction, while its zero-shot classification performance is reduced from 73.0% to 72.8%. Compared to the pruned model with $r = 1$, the model with $r = 0.5$ reduces additional 8.0% parameters at the cost of 0.2% accuracy loss. Based on the above experimental results, we set $r = 1$ for better performance recover.

### 4.6.2 THE EFFECT OF LOSS ITEMS

In this section, we analyze the effectiveness of loss items in our method. As Table 6, our method trained by $\mathcal{L}_{\text{cls}} + \mathcal{L}_{\text{patch}}$ achieves the best performance for both OpenCLIP-g and EVA-CLIP-E. Compared to our method trained by $\mathcal{L}_{\text{cls}} + \mathcal{L}_{\text{patch}}$, either the baseline with only $\mathcal{L}_{\text{cls}}$ or only $\mathcal{L}_{\text{patch}}$ leads to a significant performance degradation. For OpenCLIP-g, the baseline with only $\mathcal{L}_{\text{cls}}$ and the one with only $\mathcal{L}_{\text{cls}}$ respectively yield 0.5% and 0.9% performance loss. For EVA-CLIP-E, the baseline with only $\mathcal{L}_{\text{cls}}$ and the one with only $\mathcal{L}_{\text{cls}}$ respectively produce 0.4% and 1.0% performance degradation. We analyze the potential reason as

Table 6: The effect of loss items on average zero-shot classification accuracy. "w/o distill" denotes that the pruned model is not finetuned by knowledge distillation. The expansion ratios of all models after pruning are set to $r = 1$.

| Method | OpenCLIP-g | EVA-CLIP-E |
|---|---|---|
| original | 73.0% | 81.0% |
| w/o distill | 0.41% | 0.39% |
| $\mathcal{L}_{\text{cls}}$ | 72.5% | 80.6% |
| $\mathcal{L}_{\text{patch}}$ | 72.1% | 80.0% |
| $\mathcal{L}_{\text{cls}} + \mathcal{L}_{\text{patch}}$ | **73.0%** | **81.0%** |

follows. On the one hand, since the class token is used to predict the final output of model, training with only $\mathcal{L}_{\text{patch}}$ affects the fitting on class token for prediction. On the other hand, different from patch tokens, the class token is a parametric token, whose variance among different input data is smaller than the patch tokens. Hence, training with only $\mathcal{L}_{\text{cls}}$ is easy to cause overfitting and leads to performance degradation.

## 5 CONCLUSION

In this paper, we investigate a model pruning strategy to effectively compress large vision transformer models in a lossless manner. To this end, we focus on the parameter-intensive MLP modules in transformer architecture and propose a novel diversity-guided MLP pruning method. The proposed method prunes redundant neurons of MLP hidden layer using Gram-Schmidt algorithm, while preserving the diversity of weights to improve the recoverability of the pruned model. Then, knowledge distillation is applied to guide the performance recover of the pruned model. The experimental results on several state-of-the-art large vision transformer models demonstrate that our method accomplishes a remarkable parameters and FLOPs pruning of the given models with only negligible performance degradation. It indeed supports that our method can effectively preserve the knowledge of the original models after pruning, thus achieving effective performance recover after knowledge distillation.

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

## A  APPENDIX

In this document, we supplement additional implementation details of our method and more experimental results, including the adaption of text encoders and image classification on Swin Transformer.

## A.1 Implementation Details

For the knowledge distillation of our method, we adopt AdamW optimizer with bfloat16 precision. The learning rates of all models follows a cosine schedule for lr to min_lr, where lr = base_lr × batch_size / 256. We summarize batch_size, base_lr and min_lr in Table 7, where the setting of batch_size is based on the memory size of RTX A6000 GPUs (8 × 48G GPUs).

Table 7: The hyperparameters of our knowledge distillation method. "DDP" and "FSDP" denote distributed data parallel strategy and fully sharded data parallel strategy, respectively.

| Model | batch_size | base_lr | min_lr | weight decay | beta1 | beta2 | distribution strategy |
|---|---|---|---|---|---|---|---|
| OpenCLIP-g (ours, $r = 1$) | 224 | $1 \times 10^{-4}$ | $1 \times 10^{-6}$ | 0 | 0.90 | 0.95 | DDP |
| OpenCLIP-g (ours, $r = 2$) | 176 | $1 \times 10^{-4}$ | $1 \times 10^{-6}$ | 0 | 0.90 | 0.95 | DDP |
| OpenCLIP-G (ours, $r = 1$) | 320 | $1 \times 10^{-4}$ | $1 \times 10^{-6}$ | 0 | 0.90 | 0.95 | DDP |
| OpenCLIP-G (ours, $r = 2$) | 320 | $1 \times 10^{-4}$ | $1 \times 10^{-6}$ | 0 | 0.90 | 0.95 | DDP |
| EVA-CLIP-E (ours, $r = 1$) | 320 | $6 \times 10^{-5}$ | $1 \times 10^{-6}$ | 0 | 0.90 | 0.95 | DDP |
| EVA-CLIP-E (ours, $r = 2$) | 128 | $6 \times 10^{-5}$ | $1 \times 10^{-6}$ | 0 | 0.90 | 0.95 | DDP |
| InternVL-C (ours, $r = 1$) | 112 | $1 \times 10^{-4}$ | $1 \times 10^{-6}$ | 0 | 0.90 | 0.95 | FSDP |
| InternVL-C (ours, $r = 2$) | 80 | $1 \times 10^{-4}$ | $1 \times 10^{-6}$ | 0 | 0.90 | 0.95 | FSDP |
| EVA-CLIP-8B (ours, $r = 1$) | 128 | $6 \times 10^{-5}$ | $1 \times 10^{-6}$ | 0 | 0.90 | 0.95 | FSDP |
| EVA-CLIP-8B (ours, $r = 2$) | 128 | $6 \times 10^{-5}$ | $1 \times 10^{-6}$ | 0 | 0.90 | 0.95 | FSDP |
| DINOv2-g (ours, $r = 1$) | 512 | $7 \times 10^{-6}$ | $1 \times 10^{-7}$ | 0 | 0.90 | 0.95 | DDP |

## A.2 Algorithm

The overall algorithm is summarized as Algorithm 1.

---

**Algorithm 1** Diversity-Guided MLP Pruning

**Input:** The hidden linear weight $W_{\text{hidden}}$, bias $b_{\text{hidden}}$ and the output linear weight $W_{\text{output}}$; the target expansion ratio of the pruned MLP $r$.

**Output:** Pruned weights $W_{\text{hidden}}^{\text{pruned}}$, $W_{\text{output}}^{\text{pruned}}$ and $b_{\text{hidden}}^{\text{pruned}}$.

1: Initialize neuron set $\mathcal{J} \leftarrow \varnothing$ and $V \leftarrow W_{\text{hidden}}$;
2: **for** step = 1 to $(r \cdot M)$ **do**
3:     Select neuron $j$ with the largest $\ell_2$ norm of $\{v_i\}_{i=1}^M$ by Eq. 1;
4:     Insert selected neuron $j$ into selected neuron set $\mathcal{J}$;
5:     **for** $i = 1$ to $M$ **do**
6:         Eliminate component $v_j$ from $v_i$ by Eq. 2;
7:     **end for**
8:     **if** (step mod $N$) equals 0 **then**
9:         Obtain unselected neuron set $\mathcal{K}$ by $\overline{\mathcal{J}}$;
10:        Update $V$ by $V \leftarrow W_{\text{hidden}}[\mathcal{K}, :]$;
11:    **end if**
12: **end for**
13: Obtain $W_{\text{hidden}}^{\text{pruned}}$, $W_{\text{output}}^{\text{pruned}}$ and $b_{\text{hidden}}^{\text{pruned}}$ by Eq. 3 & 4.

---

## A.3 The Adaption of Text Encoders

In the main body of this paper, we focus on the compression of vision encoder, while the corresponding text encoder is fixed. Consequently, the fixed text encoder is not fully adapted to the compressed vision encoder, thus leading to potential performance degradation. To this end, we finetune the text encoders according to the compressed vision encoders by language-image contrastive objective (Cherti et al., 2023), where the parameters of text encoder are updated and the parameters of vision encoder are locked. All text encoders are finetuned on the training set of Flickr30K (Young et al., 2014) (image-text pairs) using AdamW optimizer with DDP strategy. The learning rate follows cosine schedule, where lr = base_lr × batch_size / 256. The hyperparameters are listed in Table 8.

As shown in Table 9, our finetuned models consistently outperform the corresponding ones without the finetuning of text encoders. Surprisingly, both EVA-CLIP-E ($r = 2$, ft) and EVA-CLIP-8B ($r = 2$, ft) surpasses their corresponding uncompressed version by a significant margin, 1.1 mean

retrieval gain. Moreover, all the finetuned models with $r = 1$ are superior to the corresponding models without the finetuning of text encoder. The above results show that the models pruned by our method can be further improved on zero-shot text retrieval and zero-shot image retrieval tasks by the finetuning of text encoder, thus supporting the effectiveness of our compression method.

Table 8: The hyperparameters for the finetuning of text encoders.

| Model | batch_size | base_lr | min_lr | weight decay | beta1 | beta2 |
|---|---|---|---|---|---|---|
| OpenCLIP-g (ours, $r = 1$) | 512 | $1 \times 10^{-6}$ | $1 \times 10^{-7}$ | 0 | 0.90 | 0.95 |
| OpenCLIP-g (ours, $r = 2$) | 512 | $1 \times 10^{-6}$ | $1 \times 10^{-7}$ | 0 | 0.90 | 0.95 |
| OpenCLIP-G (ours, $r = 1$) | 384 | $1 \times 10^{-6}$ | $1 \times 10^{-7}$ | 0 | 0.90 | 0.95 |
| OpenCLIP-G (ours, $r = 2$) | 384 | $1 \times 10^{-6}$ | $1 \times 10^{-7}$ | 0 | 0.90 | 0.95 |
| EVA-CLIP-E (ours, $r = 1$) | 2560 | $1 \times 10^{-6}$ | $1 \times 10^{-7}$ | 0 | 0.90 | 0.95 |
| EVA-CLIP-E (ours, $r = 2$) | 2240 | $1 \times 10^{-6}$ | $1 \times 10^{-7}$ | 0 | 0.90 | 0.95 |
| EVA-CLIP-8B (ours, $r = 1$) | 2048 | $1 \times 10^{-6}$ | $1 \times 10^{-7}$ | 0 | 0.90 | 0.95 |
| EVA-CLIP-8B (ours, $r = 2$) | 1600 | $1 \times 10^{-6}$ | $1 \times 10^{-7}$ | 0 | 0.90 | 0.95 |

Table 9: The performance comparison on zero-shot retrieval task of Flickr30K and COCO datasets with the adaption of text encoder. "ft" denote the text encoder of this model is finetuned to adapt the compressed vision encoder. "R@K" is short for Recall@K. "MR" is short for mean recall, which is average value of all recall metrics.

| Method | #Params | Zero-Shot Text Retrieval (text → image) | | | | | | Zero-Shot Image Retrieval (image → text) | | | | | | MR |
|---|---|---|---|---|---|---|---|---|---|---|---|---|---|---|
| | | Flickr30K | | | COCO | | | Flickr30K | | | COCO | | | |
| | | R@1 | R@5 | R@10 | R@1 | R@5 | R@10 | R@1 | R@5 | R@10 | R@1 | R@5 | R@10 | |
| OpenCLIP-g (Cherti et al., 2023) | 1.01B | 91.5 | **98.9** | 99.5 | **66.4** | 86.0 | 91.8 | 77.5 | 94.1 | 96.7 | 48.7 | 73.2 | 81.4 | 83.8 |
| OpenCLIP-g (ours, $r = 1$) | **0.48B** | 91.2 | **98.9** | 99.5 | 65.1 | 86.0 | 91.7 | 77.8 | 93.6 | 96.7 | 49.1 | 73.9 | 81.9 | 83.8 |
| OpenCLIP-g (ours, $r = 1$, ft) | **0.48B** | 91.4 | 98.6 | 99.6 | 65.2 | 85.7 | 91.4 | 78.9 | 94.4 | **97.1** | 49.2 | 74.0 | **82.5** | 84.0 |
| OpenCLIP-g (ours, $r = 2$) | 0.64B | 92.0 | **98.9** | **99.7** | 66.1 | 86.6 | 91.9 | 77.9 | 94.1 | 96.7 | 49.1 | 73.8 | 82.1 | 84.1 |
| OpenCLIP-g (ours, $r = 2$, ft) | 0.64B | **92.3** | 98.6 | 99.4 | 65.7 | **86.2** | **92.1** | **79.5** | **94.6** | **97.1** | **49.3** | **74.1** | **82.5** | **84.3** |
| OpenCLIP-G (Wortsman, 2023) | 1.84B | 92.6 | 99.4 | 99.8 | 66.9 | 87.2 | 92.8 | 79.8 | 95.1 | 97.0 | 51.3 | 74.8 | 82.9 | 85.0 |
| OpenCLIP-G (ours, $r = 1$) | **0.80B** | 91.4 | 99.2 | **99.9** | 66.2 | 87.0 | 92.5 | 79.3 | 94.8 | 97.0 | 50.7 | 75.1 | 83.0 | 84.7 |
| OpenCLIP-G (ours, $r = 1$, ft) | **0.80B** | **93.1** | 99.3 | 99.8 | 66.9 | 87.4 | 92.5 | 80.6 | 95.3 | **97.5** | 51.4 | 75.8 | 83.6 | 85.3 |
| OpenCLIP-G (ours, $r = 2$) | 1.07B | 91.6 | 99.3 | **99.9** | 66.8 | 87.4 | 92.9 | 79.8 | 95.1 | 97.0 | 51.2 | 75.3 | 83.2 | 85.0 |
| OpenCLIP-G (ours, $r = 2$, ft) | 1.07B | 92.8 | **99.5** | 99.7 | **67.9** | **87.7** | 92.9 | **81.2** | **95.6** | 97.4 | **51.8** | **75.9** | **83.7** | **85.5** |
| EVA-CLIP-E (Sun et al., 2023) | 4.35B | 94.9 | 99.3 | 99.7 | 68.8 | 87.8 | 93.0 | 78.9 | 94.4 | 97.1 | 51.0 | 74.8 | 82.7 | 85.2 |
| EVA-CLIP-E (ours, $r = 1$) | **1.24B** | 93.2 | 99.3 | 99.7 | 68.1 | 87.8 | 93.1 | 79.5 | 94.8 | 96.9 | 51.6 | 75.2 | 82.9 | 85.2 |
| EVA-CLIP-E (ours, $r = 1$, ft) | **1.24B** | 94.1 | 99.5 | 99.9 | 69.1 | 88.5 | 93.3 | 82.1 | 95.8 | 97.7 | 52.8 | **76.6** | 84.3 | 86.1 |
| EVA-CLIP-E (ours, $r = 2$) | 1.65B | 93.4 | 99.4 | 99.6 | 68.5 | 87.9 | 93.1 | 79.8 | 94.9 | 97.0 | 51.6 | 75.3 | 83.1 | 85.3 |
| EVA-CLIP-E (ours, $r = 2$, ft) | 1.65B | **94.7** | **99.7** | **100.0** | **69.3** | **88.4** | **93.7** | **82.0** | **96.1** | **97.8** | **52.9** | **76.6** | **84.5** | **86.3** |
| EVA-CLIP-8B (Sun et al., 2024) | 7.53B | 94.4 | 99.4 | 99.7 | 69.6 | 88.6 | 93.2 | 80.9 | 95.3 | 97.4 | 51.7 | 75.0 | 82.7 | 85.7 |
| EVA-CLIP-8B (ours, $r = 1$) | **3.23B** | 93.5 | 99.4 | 99.7 | 69.7 | 88.6 | 93.1 | 81.2 | 95.5 | 97.6 | 51.9 | 75.3 | 83.2 | 85.7 |
| EVA-CLIP-8B (ours, $r = 1$, ft) | **3.23B** | 94.7 | **99.6** | 99.8 | 70.5 | 88.6 | 93.4 | 81.0 | 95.8 | 97.6 | 52.2 | 76.4 | 84.3 | 86.2 |
| EVA-CLIP-8B (ours, $r = 2$) | 4.30B | 93.8 | **99.6** | 99.7 | 69.9 | 88.1 | 93.1 | 81.7 | 95.6 | 97.5 | 52.3 | 75.8 | 83.5 | 85.9 |
| EVA-CLIP-8B (ours, $r = 2$, ft) | 4.30B | **96.0** | **99.6** | 99.8 | **71.4** | **89.1** | **93.7** | **82.5** | **96.1** | **97.9** | **53.4** | **77.1** | **84.7** | **86.8** |

Furthermore, we also evaluate the finetuned models on zero-shot image classification task of various ImageNet variants and ObjectNet. The results are listed in Table 10. The models with the finetuning of text encoder almost keep the average zero-shot classification performance of the previous version. In future work, we plan to further finetune both vision encoder and text encoder for better adaption.

## A.4 IMAGE CLASSIFICATION ON SWIN TRANSFORMER

To further validate the effectiveness of our method, we evaluate our method on another Swin Transformer variant, Swin Transformer architecture (Liu et al., 2021) for supervised image classification. Specifically, we compress the Swin-B with image size of $224 \times 224$ and patch size of $16 \times 16$. Besides distillation on patch tokens, we further introduce cross entropy loss for supervised learning. The total loss function can be formalized as $\mathcal{L}_{\text{total}} = \mathcal{L}_{\text{patch}} + \lambda \cdot \mathcal{L}_{\text{xent}}$, where $\mathcal{L}_{\text{xent}}$ and $\lambda$ denote supervised cross entropy loss and the coefficient of supervised cross entropy loss, respectively. We find that finetuning the distilled model with $\lambda = 0.1$ achieves the best performance. For the optimization of the pruned model with $r = 1$ and $r = 2$, we follow a similar setting as the above CLIP models in Table A.1 and adopt DDP strategy. The difference is that we set base_lr $= 5 \times 10^{-4}$, min_lr $= 1 \times 10^{-6}$ and batch_size $= 256$.

As shown in Table 11, both Swin-B ($r = 1$) and Swin-B ($r = 2$) achieve comparable performance as the original Swin-B model, while the parameter number of Swin-B ($r = 1$) is only 51.9% of the original Swin-B model. The image throughout of Swin-B ($r = 1$) is also improved from 1331 images per second to 1698 images per second. The above experimental results also support the effectiveness of our method on Swin Transformer architecture.

Table 10: The performance comparison on zero-shot image classification tasks of various ImageNet variants and ObjectNet. $r$ denotes MLP expansion ratio after pruning. "#Params" denotes the number of vision encoder's parameters, excluding the parameters of text encoder. Throughout is averaged over 8 runs on single A6000 GPU with batch size of 256.

| Method | #Params | FLOPs | Throughout | IN-1K | IN-Adv | IN-R | IN-V2 | IN-Ske | ObjectNet | Avg. Acc. |
|---|---|---|---|---|---|---|---|---|---|---|
| OpenCLIP-g (Cherti et al., 2023) | 1.01B | 0.52T | 161.6 imgs/s | 78.5% | **60.8%** | 90.2% | 71.7% | 67.5% | 69.2% | 73.0% |
| OpenCLIP-g (ours, $r = 1$) | **0.48B** | **0.25T** | **272.1 imgs/s** | 78.5% | 60.4% | **90.3%** | 71.7% | **67.9%** | 69.2% | 73.0% |
| OpenCLIP-g (ours, $r = 1$, ft) | **0.48B** | **0.25T** | **272.1 imgs/s** | 78.5% | 60.4% | **90.3%** | 71.7% | 67.9% | 69.2% | 73.0% |
| OpenCLIP-g (ours, $r = 2$) | 0.64B | 0.33T | 225.7 imgs/s | 78.6% | 60.6% | **90.3%** | **71.9%** | **67.9%** | 69.6% | **73.2%** |
| OpenCLIP-g (ours, $r = 2$, ft) | 0.64B | 0.33T | 225.7 imgs/s | **78.7%** | 60.7% | 90.2% | 71.8% | 67.8% | **69.7%** | 73.1% |
| OpenCLIP-G (Wortsman, 2023) | 1.84B | 0.95T | 95.2 imgs/s | **80.1%** | **69.3%** | 92.1% | **73.6%** | 68.9% | 73.0% | **76.2%** |
| OpenCLIP-G (ours, $r = 1$) | **0.80B** | **0.41T** | **177.0 imgs/s** | **80.1%** | 69.1% | **92.2%** | 73.5% | 68.9% | 73.0% | 76.1% |
| OpenCLIP-G (ours, $r = 1$, ft) | **0.80B** | **0.41T** | **177.0 imgs/s** | **80.1%** | 69.0% | **92.2%** | 73.4% | 69.0% | 73.0% | 76.1% |
| OpenCLIP-G (ours, $r = 2$) | 1.07B | 0.55T | 144.9 imgs/s | 80.0% | 68.5% | **92.2%** | **73.6%** | **69.2%** | 73.0% | 76.1% |
| OpenCLIP-G (ours, $r = 2$, ft) | 1.07B | 0.55T | 144.9 imgs/s | **80.1%** | 68.3% | **92.2%** | 73.4% | **69.2%** | 72.9% | 76.0% |
| EVA-CLIP-E (Sun et al., 2023) | 4.35B | 2.23T | 46.0 imgs/s | **82.0%** | 82.1% | 94.5% | 75.7% | 71.6% | 79.6% | 80.9% |
| EVA-CLIP-E (ours, $r = 1$) | **1.24B** | **0.63T** | **139.9 imgs/s** | 81.9% | **82.3%** | **94.6%** | **75.8%** | 79.5% | 79.5% | 81.0% |
| EVA-CLIP-E (ours, $r = 1$, ft) | **1.24B** | **0.63T** | **139.9 imgs/s** | 81.9% | **82.3%** | **94.6%** | **75.8%** | 71.8% | 79.5% | 81.0% |
| EVA-CLIP-E (ours, $r = 2$) | 1.65B | 0.85T | 109.9 imgs/s | 81.9% | 82.2% | **94.6%** | **75.8%** | **72.0%** | **79.7%** | **81.1%** |
| EVA-CLIP-E (ours, $r = 2$, ft) | 1.65B | 0.85T | 109.9 imgs/s | 81.9% | **82.3%** | **94.6%** | **75.8%** | **72.0%** | **79.7%** | **81.1%** |
| EVA-CLIP-8B (Sun et al., 2024) | 7.53B | 3.86T | 26.6 imgs/s | **83.5%** | 85.2% | **95.3%** | 77.7% | **74.3%** | 81.2% | **82.9%** |
| EVA-CLIP-8B (ours, $r = 1$) | **3.23B** | **1.66T** | **54.2 imgs/s** | 83.1% | 85.1% | 95.1% | **77.8%** | 73.9% | 81.3% | 82.7% |
| EVA-CLIP-8B (ours, $r = 1$, ft) | **3.23B** | **1.66T** | **54.2 imgs/s** | 83.1% | 84.6% | 95.0% | 77.7% | 73.8% | 80.9% | 82.5% |
| EVA-CLIP-8B (ours, $r = 2$) | 4.30B | 2.21T | 43.2 imgs/s | 83.4% | **85.8%** | **95.3%** | 77.7% | 74.1% | **81.4%** | **82.9%** |
| EVA-CLIP-8B (ours, $r = 2$, ft) | 4.30B | 2.21T | 43.2 imgs/s | 83.4% | **85.8%** | **95.3%** | 77.6% | 74.1% | **81.4%** | **82.9%** |

Table 11: The performance comparison on image classification task of ImageNet-1K using Swin Transformer architecture.

| Method | #Params | FLOPs | Throughout | Accuracy (%) |
|---|---|---|---|---|
| Swin-B (Liu et al., 2021) | 87.8M | 30.3G | 1641 imgs/s | 83.4 |
| Swin-B ($r = 1$) | 45.6M | 15.5G | 2752 imgs/s | 83.3 |
| Swin-B ($r = 2$) | 59.8M | 20.4G | 2203 imgs/s | 83.4 |

## A.5 THE VISUALIZATION OF WEIGHT DIVERSITY

We evaluate weight diversity using PCA (Principal Component Analysis). Specifically, we decompose the weights into several component vectors and the variances along the corresponding component vector. The variances can be used to measure weight diversity along the corresponding component vector.

We compare the weight diversity of our method with random pruning on OpenCLIP-g for the first 12 blocks and plot the variances of different components in the following anonymous link, where components are ranked by the value of variances (1408 components for OpenCLIP-g). As show in the Figure 4, compared to random pruning, the variances of our method are closer to the ones of the original model, especially for our method of the first 12 blocks with $r = 2$. The above results indeed support that our method can well preserve the diversity of the original model.

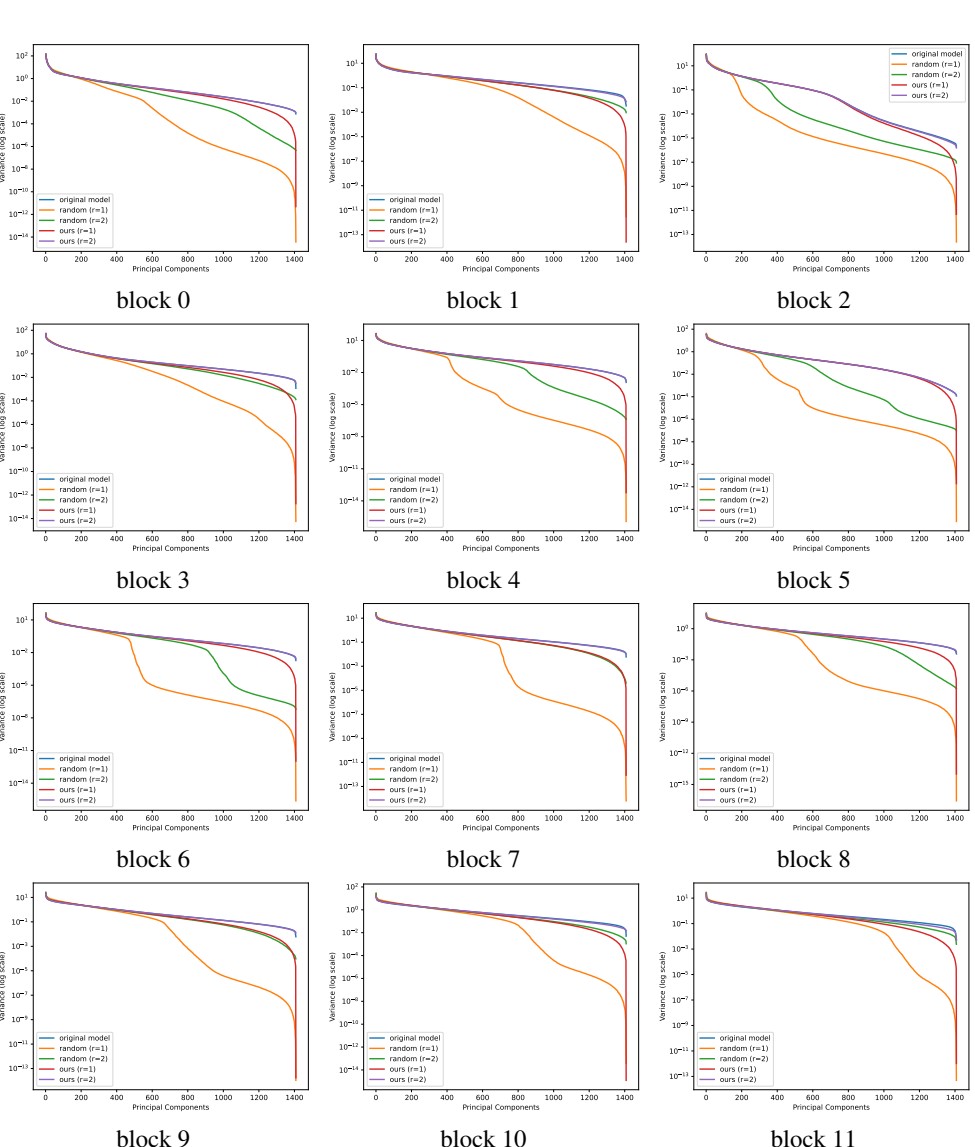

Figure 4: The component variance distribution of OpenCLIP-g model weights after pruning using PCA, where "original' model", "random (r=*)" and "ours (r=*)" denote the original model without pruning, the model with random pruning and the model pruned by our proposed method, respectively.

