# OpenReview forum: "Diversity-Guided MLP Pruning for Efficient Large Vision Transformers"
_ICLR.cc/2026/Conference — ICLR 2026 Conference Withdrawn Submission_

### Official Review · Reviewer_Yd2M · 2025-10-26

**Soundness:** 2
**Presentation:** 3
**Contribution:** 2
**Rating:** 4
**Confidence:** 4

**Summary:**

The paper proposes Diversity-Guided MLP Pruning (DGMP) for post-training pruning of vision transformers. The method prunes hidden neurons of MLP layers in transformers iteratively by ranking the neuron weight norms, while additionally applying Gram-Schmidt iteration to preserve the independent component of remaining neurons. The method is followed by a knowledge distillation stage to recover the performance. Experiments on several state-of-the-art vision transformersshow that the proposed method achieves more than 57% parameter and FLOPs reductions in a lossless manner.

**Strengths:**

1. The manuscript is easy to follow and well-organized.
2. The work shows solid results on vision transformers.

**Weaknesses:**

1. **Limited comparison with strong structured-pruning baselines.** The main concern of the paper is that there are many recent strong structured pruning baselines for general transformer pruning that are highly relevant and should be discussed or compared in dicussion or experiments, for example:
- Ma, Xinyin, Gongfan Fang, and Xinchao Wang. "Llm-pruner: On the structural pruning of large language models." Advances in neural information processing systems 36 (2023): 21702-21720.
-  Wang, Yuxin, et al. "CFSP: An Efficient Structured Pruning Framework for LLMs with Coarse-to-Fine Activation Information." Proceedings of the 31st International Conference on Computational Linguistics. 2025.
- Li, Guangyan, Yongqiang Tang, and Wensheng Zhang. "LoRAP: Transformer Sub-Layers Deserve Differentiated Structured Compression for Large Language Models." Forty-first International Conference on Machine Learning.
- Ling, Gui, Ziyang Wang, and Qingwen Liu. "Slimgpt: Layer-wise structured pruning for large language models." Advances in Neural Information Processing Systems 37 (2024): 107112-107137.
- An, Yongqi, et al. "Fluctuation-based adaptive structured pruning for large language models." Proceedings of the AAAI Conference on Artificial Intelligence. Vol. 38. No. 10. 2024.
To validate the performance of the proposed method, some of these methods should be included.
2. **Limited experients beyond vision-only cases.** Another concern is that the method is specifically designed and only tested on vision transformer. Since structured pruning is a general architecture-level technique, one would expect DGMP to also benefit other transformer modalities, such as pure text-based transformers. Even a small experiment on text-based LLM would significantly improve the paper’s general relevance and demonstrate that the proposed method is a universal principle rather than a ViT-specific heuristic.

**Questions:**

Can the authors provide detailed experiment/discussion comparison with the mentioned structured-pruning baselines?

---

> ### Author Response · Authors · 2025-12-03
> **The comparison on LLMs is **out of scope** of our paper**
>
> As claimed in the title of our paper, we **only claim the contributions** on the pruning of **vision models** on **vision tasks**, **instead of general transformer pruning**.

---

### Official Review · Reviewer_JbNE · 2025-10-27

**Soundness:** 2
**Presentation:** 2
**Contribution:** 2
**Rating:** 4
**Confidence:** 3

**Summary:**

This paper proposes Diversity-Guided MLP Pruning (DGMP) to compress large-scale Vision Transformer (ViT) models by focusing on the MLP layers, which occupy most parameters in ViTs. Motivated by the observation that redundant neurons can be represented by a combination of a few principal neurons, DGMP evaluates neuron importance using the L2 norm, then iteratively selects diverse neurons using the Gram-Schmidt process to maximize diversity. This approach avoids redundancy and improves recoverability of the pruned model without requiring gradient-based optimization or iterative fine-tuning. To restore performance, DGMP leverages knowledge distillation with both class token loss $\mathcal L_{\rm cls}$ and patch token loss $\mathcal L_{\rm patch}$ to guide the training of the pruned model. Extensive experiments show that DGMP achieves significant parameter and FLOPs reduction with minimal or no accuracy degradation. Notably, on zero-shot image classification and zero-shot retrieval tasks, the pruned models even outperform the original ones in some cases.

**Strengths:**

1. The paper is well-motivated by the insight that preserving neuron diversity improves the recoverability of pruned models.

2. Extensive experiments validate the effectiveness of the proposed method across multiple large-scale models (e.g., OpenCLIP, EVA-CLIP, InternVL-C, DINOv2) and tasks (zero-shot classification, retrieval, kNN evaluation), consistently showing strong performance with minimal accuracy drop.

3. The pruning algorithm avoids expensive gradient computations and iterative fine-tuning, making it practical and scalable for very large models.

**Weaknesses:**

1. Neuron importance is estimated based solely on $W_{\text{hidden}}$.
It would be interesting to consider whether incorporating the output-side weights $W_{\text{output}}$ could further improve the pruning strategy, as some neurons might contribute more strongly to downstream layers than their input-side connectivity would suggest.

2. Knowledge distillation may still require substantial fine-tuning.
Although the method avoids iterative pruning and gradient-based ranking, the distillation stage involves multiple epochs of training, which could be viewed as a form of fine-tuning. A clarification on how this differs from traditional pruning–fine-tuning pipelines would be helpful.

3. Unclear whether the method is specifically tailored for large-scale ViTs.
The paper emphasizes large vision transformers, but the method appears general enough to apply to smaller models as well. Including results on standard models like ViT-B or ImageNet-1K classification would help clarify its broader applicability.

4. Comparison baselines could be updated.
In Section 4.4, the pruning baselines are relatively older. Including more recent or stronger pruning baselines—especially those proposed in the last year—would strengthen the experimental comparisons.

5. Fairness of baseline application may be questionable.
The pruning baselines are applied only to the MLP layers, even for methods like NViT that are designed for global pruning. A justification or sensitivity analysis on this design choice would improve fairness and transparency.

6. Minor technical inaccuracies and typographical errors.
There are a few minor inconsistencies in the writing that could be clarified or corrected.
For example, on page 5, the sentence “Then, we select the next neuron by Table 1 and ~” seems to incorrectly reference Table 1, which presents experimental results, rather than Eq. (1), which defines the selection criterion.
Additionally, the Gram-Schmidt algorithm is repeatedly misspelled as “Gram-Schmid” throughout the paper.

**Questions:**

Refer to Weaknesses

---

> ### Author Response · Authors · 2025-12-04
>
> Thanks for your feedback. We address your concerns as follows.
>
> - **The pruning of output-side weights**
>
>   Since the skip connections in Transformer architecture, the pruning of output-side weights of MLP will change feature dimension of all Transformer blocks and **significantly change the network structure**, thus **introducing unrecoverable performance degeneration**. Our paper focuses on controllable compression of Large Vision Transformers, whose performance can be well recovered after pruning.
>
>   Moreover, output-side weights are **significantly smaller than hidden weights** (e.g.  output-side weights only contain 12.5% parameters of hidden weights for EVA-CLIP-E). Therefore, the pruning of output-side weights is not a cost-effective option.
>
> - **The differences from traditional pruning–fine-tuning pipelines**
>
>   We summarize the differences between our method and traditional pruning–fine-tuning pipelines.
>
>   - Distillation can **well exploit dark knowledge from the logits of teacher** vision models, which provides more information than image-text paired data. Hence, our method can **achieve better generalization on wider data**, when only very limited data are used for finetuning.
>   - The differences on **training dataset**. Our method only requires image data, **instead of image-text paired data** used for the finetuning of CLIP-style models. Distillation used by our method can be well adapted to various vision models.
>   - The differences on **trained model**. Our method only requires vision encoder and doesn't need text encoder. It significantly reduces the complexity of finetuning.
>
> - **The effectiveness on smaller ViT models of ImageNet-1K**
>
>   We indeed validate the effectiveness of our method on smaller ViT models of ImageNet-1K. In **Section A.4 of appendix**, we report the results on Swin Transformer models trained on ImageNet-1K. The experimental results indeed support the effectiveness of our method on smaller ViT models.
>
> - **The fairness of comparison to NViT**
>
>   Since NViT **can be only applied to the attention module of ViT after its mortification**, NViT can not be used to pretrained large vision transformer models in our paper. Hence, we only apply NViT on MLP modules of large pretrained models. We confirm that the comparison between NViT and our method is fair in the same pruning settings.
>
> - **Minor technical and typing issues**
>
>   We will fix them in the revision.

---

### Official Review · Reviewer_4Eir · 2025-11-03

**Soundness:** 2
**Presentation:** 2
**Contribution:** 2
**Rating:** 2
**Confidence:** 5

**Summary:**

This paper focuses on pruning Vision Transformer models, particularly large ViTs, to achieve a better boundary of the trade-off between model parameters and model performances. Specifically, this paper targets the high hidden dimensions (expansion ratio) in the MLP block of each Transformer layer and proposes a pruning method named DGMR inspired by Gram Schmit orthogonalization along with a distillation to recover the model capacity. Experiments on ImageNet zero-shot classification and image-text retrieval shows the effectiveness of DGMR.

**Strengths:**

1. The method seems simple and easy to implement on different types of Vision Transformer models. The proposed DGMR may have large potential on many large ViTs to slim their computation.

2. The experimental results on zero-shot classification and multimodal retrieval are significant and convincing, demonstrating the effectiveness of the method.

3. The figure is clean and cool to understand the method.

**Weaknesses:**

1. The concept of "diversity" may not be appropriate in this paper. The authors do not give a well-defined formulation of the diversity mentioned in the paper and the "guidance" of diversity is not elaborated in the method section. If the paper regards the diversity as the rank or the dimension of the linear space of the projection matrix between input and hidden neurons, there hold an implicit hypothesis that a higher linear space dimension would lead to a higher feature diversity for the hidden neurons. However, the effect of the non-linear activation function after this projection is not clear whether it could increase or decrease the diversity. The detailed investigation and analysis of this issue are also missing in the paper, which is essential to navigating the problem and justifying the proposed DGMR pruning method. This absence makes the paper less well-motivated and less generalizable.

2. Although the paper exhibits many impressive experimental results on CLIP zero-shot image classification, image-text / text-image retrieval, and k-NN representation learning, results of pruned Large ViTs on Vision Language Models (VLMs) or Multimodal Large Language Models are omitted. Since Large ViTs usually serve the vision encoders in VLMs, whose performance is critical to many complicated VLM tasks, therefore they have larger impacts.

3. The distillation after pruning is a common practice among pruning methods, which does not contribute enough to the novelty. Table 6 shows that the model would fail to adapt to the tasks after pruning MLP neurons using DGMR without the post-pruning distillation stage. This result may imply that the distillation process is the key component in the whole pipeline of the method, which has not been fully investigated. The author should provide an analysis of hyperparameters including distillation epochs, substituting to different teachers with the same hidden dimensions, or just fine-tuning on the image-text pertaining datasets. If the paper insists on distillation as a major contribution, there should be more explanations and analyses of why it works best for the proposed DGMR method.

4. The abstract and introduction writing needs some improvements, e.g., the necessity of Gram Schmit strategy is not well established in the abstract, the logic of the introduction section is not clear, filled with many "due to" and "afterward", which makes the paper hard to follow.

**Questions:**

I have no other questions.

---

> ### Author Response · Authors · 2025-11-22
> **Serious Concerns**
>
> The review has, unfortunately, largely ignored the main contribution and outstanding performance of our proposed method, yet posted with a confidence of `Confidence: 5: You are absolutely certain about your assessment`.
>
> - The reviewer suggests a **significantly expanded scope for the evaluation benchmarks**. However, we would like to respectfully note that this is a **conference submission** aimed at the **rapid dissemination of novel ideas**, **not a comprehensive monograph**. Furthermore, we are strictly constrained by **ICLR's maximum limit of 10 pages**.
>
> - The reviewer appears to **confuse our main contribution (diversity-guided MLP pruning, DGMP) with distillation, focusing their critique on the latter**.
>
>   However, our key insight is that DGMP produces models that are uniquely suited for performance recovery via distillation. We extensively demonstrate that **pruned models from other baselines cannot effectively recover performance under the same conditions**. Thus, the contribution lies in the specific pruning strategy that facilitates this recovery, not the distillation technique itself.
>
> - Focus on several trivial points that don't affect the understanding of this paper.

---

### Official Review · Reviewer_ToHJ · 2025-11-10

**Soundness:** 3
**Presentation:** 3
**Contribution:** 2
**Rating:** 4
**Confidence:** 3

**Summary:**

This paper proposes parameter pruning for the MLP module, which dominates the model parameters in the transformer architecture. This pruning significantly reduces the number of parameters while incurring only a negligible performance loss. Specifically, the authors propose a Diversity-Guided MLP Pruning method, employing a Gram-Schmidt weight pruning strategy to eliminate redundant neurons in the MLP hidden layers and maintain weight diversity to improve performance recovery during the pruning process. Finally, extensive experiments validate the proposed method.

**Strengths:**

The paper clearly and logically presents its innovative aspects to the reader.

**Weaknesses:**

The key innovation of this paper lies in the Gram-Schmidt weight pruning strategy. However, the analysis of this part in the experimental evaluation is still not comprehensive enough.

**Questions:**

1. The paper points out that the weight parameters of the MLP module in the transformer architecture model are dominant. However, the experiments only validated the method on ViT. To demonstrate the generality of the DGMP method, larger transformer architecture models such as LLama-7B and DeepSeek can be used for validation.
2. Although the authors claim that their approach does not require iterative fine-tuning, the knowledge distillation stage still relies on the computationally expensive teacher model. It would be valuable to further analyze and compare the influence of iterative fine-tuning and knowledge distillation on the DGMP pruning framework, particularly in terms of convergence efficiency, performance recovery, and the overall trade-off between computational cost and model fidelity.
3. In the ablation experiments, the pruning comparison method presented in the paper is somewhat outdated, and the accuracy evaluation dimensions are relatively singular.

---

> ### Author Response · Authors · 2025-12-03
>
> Thanks for your feedback. We address your concerns as below.
>
> - **Validation on LLMs, such as  LLama-7B and DeepSeek**.
>
>   The compression of LLMs is another interesting topic. However, our paper **only claims the contributions on vision transformers**, not including language transformers. We are researching on the exploration of our method on LLMs, which is more time-consuming.
>
> - **The analysis of iterative fine-tuning efficiency**.
>
>   The efficiency iterative fine-tuning highly depends on hyper-parameters, such as the percentage of parameters for each pruning stage. However, our method can well recover the performance of pruned model after 10-epoch distillation. For example, EVA-CLIP-E (r=1) only takes 24.3 hours on 8 A6000 GPUs, which is affordable for academic researchers.

---

### Note · Authors · 2026-02-05

I have read and agree with the venue's withdrawal policy on behalf of myself and my co-authors.

---

### Meta-Review · Area_Chair_ACTf · 2026-01-07

**Summary:**

All reviewers acknowledge the clarity of the writing and the strong motivation of the paper. Reviewers 4Eir, JbNE, and Yd2M further recognize the sound design of the proposed method and its strong experimental results on vision transformers. The authors have adequately addressed the concerns raised by Reviewer JbNE regarding the pruning of output-side weights, the distinctions from traditional pruning-and-fine-tuning pipelines, the effectiveness on smaller ViT models trained on ImageNet-1K, and the fairness of comparisons with NViT. However, as noted by multiple reviewers, the proposed method is general in nature and potentially applicable to general transformer architectures beyond vision transformers. In this context, several state-of-the-art structured pruning methods designed for general transformers were neither sufficiently compared against nor properly discussed in the manuscript. This limitation was not satisfactorily addressed during the rebuttal. Considering these unresolved concerns, we do not recommend acceptance at this stage, and suggest that the manuscript be further polished to incorporate a more comprehensive comparison and discussion to resolve the identified issues.

**Reviewer Concerns:**

Part of Reviewer JbNE are addressed, such as explanation of the pruning of output-side weights, the effectiveness on smaller ViT models of ImageNet-1K, the differences from traditional pruning–fine-tuning pipelines, and the explanation of fairness issue of experimental settings. However, JbNE did request the authors to compare more recent baselines but the authors did not respond to the question. For other reviewers, the authors claimed their methods focus on vision transformers and thus did not respond to the concerns to be compared with other state-of-the-art structured pruning methods which are applied on general transformers, not vision transformers.

**Reviewer Scores:**

In general, most of the reviewers have the concerns how the proposed DGMP method is compared with other state-of-the-art structured pruning methods which could be applied beyond vision-only transformer. For example, Reviewer Yd2M provides several references and asks the authors to either discuss or provide a small comparison. All the reviewers indicate this insufficiency for experimental comparisons. The authors explain that they mainly focus on vision transformers. There is no intersection between the review and the rebuttal. In this case, no reviewers will change their ratings.

---

### Decision · Program_Chairs · 2026-01-26

Reject